# Plectin in Cancer: From Biomarker to Therapeutic Target

**DOI:** 10.3390/cells10092246

**Published:** 2021-08-30

**Authors:** Samantha M. Perez, Lindsey T. Brinton, Kimberly A. Kelly

**Affiliations:** 1Department of Biomedical Engineering, University of Virginia, Charlottesville, VA 22908, USA; smp7ya@virginia.edu; 2ZielBio, Inc., Charlottesville, VA 22903, USA

**Keywords:** plectin, biomarker, imaging agent, drug delivery, therapeutic

## Abstract

The cytolinker and scaffolding protein, plectin, has emerged as a potent driver of malignant hallmarks in many human cancers due to its involvement in various cellular activities contributing to tumorigenesis, including cancer cell proliferation, adhesion, migration, invasion, and signal transduction. Evidence shows that beyond plectin’s diverse protein interactome, its cancer-specific mislocalization to the cell surface enables its function as a potent oncoprotein. As such, therapeutic targeting of plectin, its protein interactors, and, in particular, cancer-specific plectin (CSP) presents an attractive opportunity to impede carcinogenesis directly. Here, we report on plectin’s differential gene and protein expression in cancer, explore its mutational profile, and discuss the current understanding of plectin’s and CSP’s biological function in cancer. Moreover, we review the landscape of plectin as a prognostic marker, diagnostic biomarker, and target for imaging and therapeutic modalities. We highlight how, beyond their respective biological importance, plectin’s common overexpression in cancer and CSP’s cancer-specific bioavailability underscore their potential as high-value druggable targets. We discuss how recent evidence of the potent anti-cancer effects of CSP therapeutic targeting opens the door for cell-surface mislocalized proteins as novel therapeutic targets.

## 1. Introduction

First identified over four decades ago, plectin is a 500 kDa protein commonly expressed in mammalian tissues and cell types [1]. It consists of an actin-binding domain, a plakin domain, a central coiled-coil rod domain, and a plakin repeat domain, giving rise to at least 12 different isoforms by alternative splicing; as a result, plectin displays variable intracellular localization dependent on cell type (e.g., hemidesmosomes, focal adhesions, nucleus, ER membrane, mitochondria) [1,2,3,4,5,6]. Furthermore, its diversity has resulted in plectin playing an important multifunctional role in cellular organization and signal transduction. It acts as a cytolinker that binds and stabilizes membrane and cytoskeletal proteins, including microtubules (MTs), actin microfilaments, and intermediate filaments (IFs) [3]. Moreover, plectin is a scaffolding protein known to bind to the receptor for activated C kinase 1 (RACK1), thus modulating protein kinase C (PKC) signaling pathways, and it has been shown to interact with phosphatidylinositol-4,5-biphosphate (PIP2) [7], integrin α6β4 [8], and calmodulin [9], among others.

Beyond normal physiology, multiple studies have emerged implicating plectin as a pro-tumorigenic regulator of cancer cell proliferation, migration, and invasion [10,11,12]. In particular, plectin’s cancer-specific cell surface mislocalization has revealed plectin expression as a biomarker or prognostic indicator in several cancers, including pancreatic, ovarian, lung, prostate, and head and neck cancer [10,11,13,14,15,16,17]. To this end, multiple groups have successfully leveraged cancer-specific plectin’s (CSP) abundant and bioavailable expression to guide imaging agents and drug delivery systems [13,15,18,19,20,21,22,23,24]. Moreover, recent studies have revealed the anti-cancer effect of direct therapeutic targeting of CSP and plectin, opening new avenues of research into CSP’s and plectin’s role in cancer [25,26,27]. Here, we review the current understanding of plectin’s critical role in cancer biology, its diagnostic capabilities, and its therapeutic potential, all of which underscore its far-reaching biologic significance and clinical utility.

## 2. Plectin Expression in Cancer

### 2.1. Plectin Cell Surface Mislocalization in Cancer

In 2008, Kelly et al. used a phage-based functional proteomic approach to identify plectin as a specific and abundant cell surface target in pancreatic ductal carcinoma (PDAC) that remains cytoplasmic in healthy tissue [13]. Since then, flow cytometry, immunogold transmission electron microscopy, and plectin-binding assays using PDAC cell lines have conclusively demonstrated CSP expression on malignant cells (Bx.Pc3, L3.6pl, Panc-1), while normal pancreas cells (HPDE) are null for CSP [12]. Furthermore, multiple groups have reported the success of CSP-targeting imaging agents and drug delivery systems in selectively identifying PDAC cells and tumors [18,19,20,21,22,23,24]. For example, Konkalmatt et al. demonstrated how a CSP-targeting AAV2 virus bound to and accumulated in PDAC tumors whereas control healthy tissues such as pancreas, liver, and spleen were devoid of uptake (Figure 1) [18]. Wild-type virus, in contrast, showed predominant liver accumulation with minimal tumor uptake [18]. These results emphasize the selectivity of CSP expression for tumors over healthy tissues. In breast cancer mouse models with MDA-MB-231 tumors, a CSP-targeting drug delivery system specifically bound to tumors and enhanced drug efficacy [28]. Western blot analysis of the membrane fraction validated MDA-MD-231 cells as CSP-positive [29]. In ovarian cancer, CSP-targeting liposomes selectively recognize and bind to ovarian cancer cells in vitro and in vivo [15]. Moreover, immunoblot detection of plectin in the cell surface fraction of ovarian cancer cell lines (SKOV3 and OVCAR8) showed high expression compared to healthy human fallopian tube epithelial cells (FT132) [15]. In lung cancer, CSP was revealed as a cell-surface cancer-stem-cell biomarker via an unbiased peptoid combinatorial cell screen [16]. This was further validated by mass spectrometry and confirmed by competition binding assays with the CSP-targeting peptoid. Moreover, pull-down assays and Western blot analysis of cell-surface proteins revealed that NSCLC cell lines H358, H1693, H460, and H1975 have CSP-positive subpopulations [16].

Collectively, CSP-positive tumors, including pancreatic, lung, ovarian, and breast cancer, account for over 3 million annual deaths worldwide, making insights into CSP highly clinically relevant [30]. Furthermore, CSP’s diagnostic and therapeutic potential underscores the need to explore CSP status in other cancers, especially those lacking reliable biomarkers. To this end, Reynolds et al. performed plectin IHC staining on human cancer tissue microarrays [17]. They validated pancreatic, ovarian, and lung cancer tissue as having strong plectin membrane staining, consistent with their CSP-positive status [17]. Strikingly, they revealed that other cancers have significant membranous staining: colorectal adenocarcinoma, bile duct cholangiocarcinoma, head and neck squamous cell carcinoma, and intestinal-type stomach cancer [17]. These results suggest that CSP could serve as a biomarker or cancer-specific antigen for targeted therapies in multiple malignancies.

### 2.2. Plectin Is a Cancer Biomarker

Beyond its differential localization, plectin has been widely characterized as commonly overexpressed across multiple cancers, often demonstrating uniquely high expression levels compared to similarly presenting diseases and healthy tissue. Among these, increased plectin expression in PDAC has been widely studied. Invasive pancreatic cancer arises from precursor lesions of which there are two subtypes: pancreatic intraepithelial neoplasia (PanIN) and intraductal papillary mucinous neoplasms (IPMN) [31]. Plectin IHC staining of human tissues revealed that a small percentage of early lesions (PanIN I/II) were positive for plectin expression (0–3.85%), while 60% of PanIN III lesions, considered carcinoma in situ, were positive [32]. Bausch et al. reported the sensitivity and specificity of plectin for distinguishing PanIN III and PDAC from benign and low-grade PanIN lesions to be 87% and 98%, respectively [32]. Furthermore, IHC of metastatic deposits revealed that 100% of the analyzed tissues (liver, lymph nodes, and peritoneum) maintained high plectin expression [32].

Plectin has also proved a distinguishing feature of PDAC compared to chronic pancreatitis (CP), which often has a similar clinical presentation. Plectin IHC staining by Bausch et al. demonstrated that 100% of the human PDAC tissues analyzed were positive for strong membranous and cytoplasmic plectin [32]. In comparison, 100% of the benign pancreas and chronic pancreatitis (CP) tissues were negative for plectin expression [32]. Here, plectin’s differential expression between malignant and benign tissue enables differential diagnosis when clinical signs and symptoms cannot discern PDAC from CP [33]. Furthermore, PDAC can emerge in the backdrop of CP; thus, biomarkers that can distinguish between both could help improve detection and staging.

Emerging evidence has revealed that plectin overexpression is also characteristic of several head and neck cancers. This is noteworthy because head and neck squamous cell carcinoma (HNSCC), which includes cancers originating from the oral cavity, pharynx, larynx, and sinonasal tract, is the sixth most common cancer worldwide and is expected to continue to increase in incidence [34]. Using surgically resected HNSCC tissue, Katada et al. implemented 2D-DIGE, IHC analysis, and Western blotting to demonstrate that plectin is overexpressed in HNSCC compared to adjacent non-malignant tissue [10]. Moreover, proteomic analysis of laser-capture microdissected oral squamous cell carcinoma (OSCC) tissue revealed plectin overexpression in OSCC compared to healthy oral mucosas [35]. In accordance, plectin IHC and qPCR also demonstrated a significantly higher abundance of plectin in OSCC tumor tissue compared to normal epithelium [36]. Plectin IHC analysis by Rikardsen et al. also demonstrated strong, predominantly membranous, plectin staining in OSCC tissue but only faint staining in normal tongue mucosa [37]. In sinonasal squamous cell carcinoma, plectin protein expression was also significantly upregulated in malignant tissue compared to premalignant sinonasal inverted papilloma [38].

An expanded plectin IHC analysis by Bausch et al. revealed that esophageal, stomach, and lung cancers also show high differential expression compared to non-malignant tissue, suggesting that plectin could serve as a biomarker in other cancers [32]. Moreover, several independent reports have emerged, revealing differential plectin expression in ovarian, colon, endometrial, lung, and prostate, among others [10,11,15,17,39,40,41]. IHC analysis of patient specimens demonstrated strong plectin expression (>80%) at the cell membrane of serous, clear cell, and poorly differentiated ovarian cancer [15]. In comparison, low-grade tumors such as serous cystadenoma and mucinous cancers presented moderate staining of predominantly cytosolic plectin expression (>70%), while healthy tissue demonstrated weak staining [15,17]. In endometrial cancer, plectin IHC analysis revealed strong plectin expression in malignant tissues with and without lymph node metastasis but only faint staining in normal tissue [40]. Similarly, IHC analysis of patient samples in colon cancer demonstrated increased expression in colorectal adenocarcinoma and locally invasive nests compared to normal tissue [41]. In prostate cancer, IHC analysis demonstrated significantly higher plectin staining in prostate cancer tissues and lymph node metastases when compared to patient-matched benign prostate tissue and cancer-free lymph node tissues, respectively [11]. This observation is consistent with previous reports of differential plectin expression in prostate cancer [39,42]. A quantitative proteomic analysis of esophageal squamous cell carcinoma (ESCC) tissue identified plectin as overexpressed in malignant tissue compared to adjacent normal epithelium [43]. IHC analysis further demonstrated plectin overexpression in ESCC in contrast to faint staining in normal tissue [43]. Likewise, a proteomic and IHC analysis of tumors from a mouse fibrosarcoma model revealed plectin to be abundantly expressed in malignant tissue [44,45].

In contrast to the numerous reports of plectin’s overexpression in cancer, differing observations have been made in hepatocellular carcinoma (HCC) and carcinoma of the skin, highlighting the importance of elucidating tissue-specific or context-dependent exceptions. In HCC, IHC staining and Western blot analysis have demonstrated that plectin is downregulated in patient tumor tissue compared to healthy liver tissue [46,47,48,49,50]. However, quantitative phosphoproteome analyses comparing HCC patients and healthy samples have identified plectin phosphorylated at Ser4253 as a potential HCC phospho-biomarker [51,52]. In skin cancer, IHC analysis has revealed decreased plectin expression in basal cell carcinoma of the skin and, to a smaller extent, in squamous cell carcinoma and in situ carcinoma compared to adjacent normal tissue [53]. Due to these observations being limited by small cohorts or based on representative images without pathologist scoring, a large retrospective study analyzing plectin status in different HCC and skin cancer subtypes could provide further insight.

### 2.3. Plectin Mutations in Cancer

Although mutations in plectin have been linked to epidermolysis bullosa simplex, muscular dystrophy, pyloric atresia, and several central nervous system malignancies, not much is known regarding plectin genetic defects in cancer [54]. Genetic mutations and alterations are canonically thought to play a critical role in carcinogenesis. To this end, using publicly available databases cBioPortal and Oncomine, Harryman et al. revealed plectin as the most commonly altered gene across 12 epithelial cancers compared to other essential genes for laminin-binding integrin adhesion, including integrin β4, integrin α3, laminin β3 chain, and nesprin 3 [55], although it stands to reason that proteins such as the massive 500 kDa plectin may tend towards a larger number of altered genes due to their larger size. To further examine plectin’s mutation profile, we analyzed 10,967 samples spanning 32 The Cancer Genome Atlas (TCGA) cancer types from their PanCancer Atlas datasets using cBioPortal, of which 5.1% had somatic mutations (Figure 2A) [56,57]. Strikingly, of the 800 plectin mutation sites identified, all were passenger mutations with no known significance. Moreover, plectin was found to be altered in 11% (1158/10,950) of TCGA pan-cancer samples, with amplification and missense mutations being the most common genetic alterations (Figure 2B). This was reflected across different cancer types with alteration frequencies spanning 0–27% (Figure 2C). Interestingly, there was a predominant pattern of amplification in ovarian (26%), esophageal (12.6%), and pancreatic (8.1%) cancer, which coincides with previous reports of plectin overexpression in these malignant tissues [12,15,43]. Identified plectin mutations were sporadic, with no clear mutation hotspot, suggesting that mutation is likely not a major driver of plectin’s differential role in cancer. However, additional work is required to understand if the additive effects of plectin mutations and copy number alterations could lead to a pro-tumorigenic phenotype or explain the heterogenous role of plectin in different cancers.

Next, we assessed plectin’s differential gene expression between tumor and healthy tissues across different cancer types using TNMplot [58]. Impressively, plectin expression was higher across multiple tumor samples, including esophageal, ovarian, pancreatic, and stomach carcinomas, all of which have been previously reported as having upregulated plectin protein expression (Figure 2D) [14,15,17,43]. Moreover, consistent with previous IHC analysis, plectin gene expression was lower in malignant skin tissue than in healthy tissue [53]. Interestingly, plectin was also significantly overexpressed in liver hepatocellular carcinoma, suggesting nuance in the interpretation of IHC results, demonstrating decreased plectin expression in tumor tissue [46,47,48,49,50]. Similarly, plectin mRNA expression was significantly lower in colon carcinoma compared to normal tissue, while previous plectin IHC analysis demonstrated abundant expression in malignant tissue [41]. Thus, further research is required to explore how post-transcriptional modifications, cellular processes, and environmental cues modulate plectin mRNA and protein activity.

One such active area of study is the effects of epigenetic changes. Despite their critical role in driving carcinogenesis, we have a minimal understanding of how aberrant methylation or microRNA (miR) regulation affects plectin expression in cancer. miR activity associated with mutant p53 expression has been linked to plectin [59]. Mutant p53 is present in >50% of cancers and is associated with multiple pro-tumor mechanisms, including increased proliferation, invasion/migration, and metastasis, among others. Mutant p53 is regulated and mediated in part by miRs [60]. In particular, miR-661 has been shown to have anti-tumor effects in p53 wild-type cancers yet pro-tumor effects in p53 mutant cancers [59]. In addition to p53, miR-661 has also been found to interact with other genes in pro-tumor ways. For example, miR-661 promotes invasion and metastasis by inhibiting RB1 in NSCLC and increasing cell proliferation by suppressing INPP5J in ovarian cancer [61,62]. miR-661 resides within an intron of the PLEC1 gene, suggesting that the diverse biological effects of miR-661 could be intertwined with those of plectin [59]. Interestingly, plectin was also found to interact with RNA-binding protein, fused in sarcoma (FUS), which has been implicated in miR regulation [63]. The absence of plectin altered FUS’ subcellular distribution in fibrosarcoma HT1080 cells [63]. On a more global scale, in a genome-wide DNA methylation study comparing tissue samples from premalignant sinonasal inverted papilloma and sinonasal squamous cell carcinoma, plectin was revealed to be significantly hypermethylated and overexpressed in malignant tissue [38]. These studies suggest that an expanded evaluation of how epigenetic regulation and miRs affect plectin expression and functional outcomes across different cancer types is warranted.

## 3. Plectin’s Role in Cancer

An increasing number of reports have implicated plectin as a regulator of malignant phenotypes and cross-talk with the tumor microenvironment, in part due to its function as a cytoskeletal linker and signaling scaffold. Plectin has been shown to interact with RACK1, resulting in the regulation of PKC signaling and the modulation of the proliferative MAPK/ERK pathway [64,65]. Moreover, ablation of plectin in fibroblasts results in decreased Src and FAK signaling, leading to inhibited cell migration [3,7]. As a regulator of actin filament dynamics, plectin affects Rho/Rac/cdc42 signaling [7]. Additional plectin-interacting molecules involved in signal transduction include proto-oncogene Fer, calmodulin, PIP2, nesprin-3, and integrin β4 [3]. Moreover, several novel interacting partners have been identified (e.g., BRCA2, SNRPA1, RON, periplakin, Dlc1) in the context of cancer [66,67,68,69,70]. Plectin’s vast and diverse protein interactors enable plectin to regulate several cellular processes, including cell proliferation, survival, migration, and invasion (Figure 3).

### 3.1. Plectin Is a Regulator of Cancer Cell Survival and Proliferation

Plectin is involved in various processes informing the survival of malignant cells. For example, when RasV12-transformed cells are surrounded by normal cells, plectin has been shown to form a complex with paxillin and epithelial protein lost in neoplasm (EPLIN), which induces α-tubulin acetylation and microtubule rearrangement, critical steps in the apical extrusion of transformed cells from the epithelium [71,72]. Although additional studies are required to understand plectin’s importance in this cancer-preventive mechanism, this observation does suggest that increased activation of the plectin complex could serve as an anti-cancer strategy for eradicating malignant cells at the initial stage of carcinogenesis. Alternatively, plectin was identified as an interacting partner of breast cancer susceptibility protein (BRCA2), which plays an essential role in DNA damage repair and centrosome duplication [66]. During the S phase, BRCA2 associates with the centrosome and interacts with plectin to regulate centrosome localization [66]. During the M phase, plectin is phosphorylated by cyclin-dependent kinase 1/cyclin B kinase (CDK1/CycB), abolishing plectin’s crosslinking to IFs and prompting plectin rearrangement and perinuclear localization (Figure 3AI) [66,73].

Further support that plectin regulates cancer cell survival comes from loss of function studies. Knockdown of plectin was shown to induce displacement of centrosome and nuclear abnormalities, suggesting that plectin misexpression could play a part in genomic instability and, consequently, contribute to cancer development [66]. Moreover, a plectin-binding nanopeptide induced cell death in cells arrested in the G1/S phase transition in vitro and in vivo in breast cancer [74,75]. The nanopeptide binding to plectin potentially triggers cytoskeletal organization, resulting in cell death [75]. However, it remains to be elucidated whether plectin is necessary for nanopeptide-mediated cell death and, given that the nanopeptide is a natural degradation product of cyclin D2, whether its interaction with plectin is solely responsible for the cell death. Interestingly, in CD95- or tumor necrosis factor (TNF)-induced apoptosis of human breast cancer cell line MCF7, plectin was shown to be an early binding partner of active caspase 8, which results in cleavage of plectin [76]. Cleavage of plectin is thought to trigger actin depolymerization, potentially initiating the cytoskeletal reorganization typical of apoptosis (Figure 3AII) [76]. Whether induced cleavage of plectin could better prime cells for cell death remains to be elucidated.

Plectin plays a critical role in facilitating cell proliferation and tumor growth across different tumor types (Figure 3AIII). In PDAC cell lines L3.6pl, Bx.Pc3, and Panc-1, plectin knockdown by shRNA resulted in a significant decrease in proliferation, which was rescued by selective expression of plectin 1a or plectin 1f isoforms [12]. Consistently, in orthotopic PDAC models with immunocompromised and immunocompetent mice, plectin-knockdown tumors have demonstrated reduced tumor growth [12]. Similarly, in prostate cancer, plectin-knockdown cells showed decreased cell growth in vitro and inhibited tumor growth in vivo [11,39]. Moreover, in HNSCC, the ablation of plectin by siRNA suppressed cancer cell proliferation and reduced phosphorylation of ERK1/2 signaling, potentially resulting from plectin’s interaction with integrin β4 [10]. Meanwhile, the knockdown of plectin in NSCLC cells reduced clonogenicity [16].

In contrast, in mouse epidermal tumor-initiating cells, depletion of plectin by shRNA was shown to increase tumor growth; however, these observations are limited by 70% of the plectin shRNA tumors, demonstrating re-expression of plectin in 30% of its cell population [77]. The authors also note that integrin β4 can act as either a tumor suppressor or oncoprotein, with plectin’s recruitment to the plasma membrane required for integrin β4’s tumor-suppressive effects in the absence of oncogenic RAS [77]. Additional studies are needed to elucidate if the dual role of integrin β4 is related to the suppression of a plectin-dependent anti-proliferative signal in skin cancer.

### 3.2. Plectin Modulates Cancer Cell Migration, Invasion, and Metastatic Potential

Cytolinker plectin plays a crucial role in modulating IF organization and actin cytoskeleton dynamics (Figure 3AI,AII,B). It has also been shown to couple IFs to focal adhesions [74,75,78]. Moreover, in human osteosarcoma cell U2OS, plectin was revealed to be required for the functional interaction between vimentin IFs and actin stress fibers, a critical interplay for cell morphogenesis [79]. In accordance, plectin has been identified as part of actin-rich protrusions such as podosomes and invadopodia. During epithelial–mesenchymal transition (EMT), plectin has been shown to dissociate from the cytoplasmic tail of integrin β4, resulting in hemidesmosome disassembly [80]. Afterward, in invasive bladder cancer cells, plectin forms a complex with vimentin IFs and F-actin at the base of invadopodia [81]. Ablation of plectin prevents the anchorage of vimentin IFs to invadopodia, thus significantly inhibiting the cell’s ability for invadopodia formation and in vitro invasion and extravasation [81]. In vivo, tail-vein injection of plectin knockdown bladder cancer cells resulted in a significant reduction of metastases compared to the administration of cancer cells with rescued plectin and endogenous plectin expression [81]. Alternatively, in OSCC, plectin was localized to hemidesmosomes and podosomes in non-invasive cells, potentially aiding in stabilization. Plectin was not localized to invadopodia in invasive OSCC cells, instead associating with cytoplasmic fibrils [82,83]. At podosomes in colon cancer SW480 cells, the plectin-1k isoform co-localizes with integrin subunits α3, α6, β1, and β4 as well as N-WASP, cortactin, and dynamin [84]. Knockdown of plectin by siRNA impaired podosome formation, which was rescued by selective expression of plectin-1k isoform [84]. These results suggest that in OSCC, plectin could play a differential role as invadopodia replace podosomes in EMT [82]. Moreover, in OSCC cell line AW13516, vimentin knockdown clones demonstrated upregulated plectin protein levels and enhanced plectin–integrin β4 co-localization at the cell membrane, suggesting that vimentin could be a component of regulating plectin turnover and intracellular localization [85]. Furthermore, plectin and integrin β4 were co-immunoprecipitated with vimentin, suggesting plectin could link integrin β4 and vimentin [85].

Plectin’s stabilization of actin-rich protrusive structures has resulted in plectin being a critical regulator of cancer cell migration and invasion in cancer cells. In accordance, plectin is upregulated in highly metastatic and invasive cancer cells (Figure 3B). Plectin was found to be upregulated in a high-metastatic subpopulation of the bladder cancer cell line, KK-47 [81]. Western blot analysis revealed increased plectin expression in additional invasive bladder cancer cell lines, T24, BOY, and YTS-1, compared to non-invasive bladder cells, RT4 and 5637 [81]. This was replicated in primary cells from patients with invasive bladder cancer [81]. Moreover, proteomic analysis revealed increased plectin abundance in the more metastatic CL-15 lung adenocarcinoma cell line compared to the CL1-0 cell line [86]. In colon cancer, plectin expression was significantly increased in higher-grade SW480 cells than lower-grade HT29 cells [41]. Suppression of plectin by siRNA disrupted actin dynamics and inhibited adhesion, migration, and invasion [84]. Proteomic comparison of the prostate cancer cell line PC3 and a metastatic derivative PC3 cell line demonstrated a robust increase in the abundance of plectin and vimentin, a key EMT marker, in the more aggressive line [39,87]. In vitro, plectin-knockdown prostate cancer cells demonstrate decreased migration and invasion [11,39]. In vivo, knockdown of plectin resulted in reduced prostate tumor growth and metastatic burden [11]. In OSCC cell line AW13516, loss of plectin resulted in decreased cell migration and invasion, disruption of actin organization, and reduced filopodia length [88]. In a subcutaneous xenograft model, mice bearing OSCC plectin-knockdown tumors demonstrated decrease tumor growth [88]. Similarly, ablation of plectin impaired cell migration in HNSCC, NSCLC, and breast cancer cell lines [10,16,69]. In PDAC cells, Shin et al. demonstrated that loss of plectin significantly inhibited cancer cell migration and invasion across different PDAC cell lines (BxPc3, L3.6pl, and Panc-1) [12]. Moreover, in immunocompromised and immunocompetent orthotopic mouse models, plectin-knockdown tumors demonstrated reduced metastatic burden [12]. However, contrary to Shin et al., using the same PDAC cell line, BxPc3, Yu et al. showed that loss of plectin by shRNA resulted in increased PDAC cell migration in vitro [12,68]. While Yu et al. did not implement mouse models, this divergence in vitro could be due to differences in experimental conditions (Shin et al.: transwell-migration assay; Yu et al.: scratch assay) or efficiency of plectin knockdown.

In vitro suppression of plectin in cervical cancer HeLa (Chang liver) cells resulted in reduced cytokeratin 18 expression, hemidesmosome disassembly, IF disorganization, increased actin-rich stress fibers, and altered cellular morphology [47,48,89]. However, in contrast to other cancers, plectin deficiency in this cervical cancer model increased FAK and Rac1-GTPase activity, cell motility, and enhanced cell migration [47,90]. Similarly, hepatoma cell lines with lower plectin expression, PLC/PRF/5, and HepG2 demonstrated higher migration rates than cells with higher plectin expression, the Chang liver cells [90]. Additional studies could validate these observations in vivo and explore if plectin’s role in informing cell motility and the risk of metastasis is tissue-specific or context-dependent.

Several direct and indirect cancer-associated plectin interacting partners have emerged, which could provide further insight into the diversity of cellular processes and signaling networks involving plectin in cancer (Table 1). Additional work is required to understand the functional significance of these novel interactions.

### 3.3. Plectin Cross-Talks with the Tumor Microenvironment and Immune System

Given that an impaired immune system is a hallmark of solid tumors, there is great interest in identifying key regulators to inform appropriate therapeutic strategies [91]. Strikingly, the selective inhibition of plectin and CSP with a metallodrug, plecstatin-1, induced an in vitro immunogenic cell death signature [25]. Blocking plectin and CSP resulted in the secretion of ATP, the release of high mobility group box-1 (HMGB-1), and increased translocation of danger-associated molecular patterns (DAMPs) such as calreticulin, HSP90, and HSP70 to the plasma membrane [25]. Although additional studies are required to evaluate plectin’s immunogenic role in vivo, this report demonstrates that inhibition of plectin could have immunogenic consequences. Plectin has also been implicated in immune cell motility [92]. Plectin isoform 1-specific knockout mice resulted in reduced motility of dermal fibroblasts and T-cells in vitro as well as impaired infiltration of macrophages and T-cells during wound healing in vivo [92]. Although the mechanisms underpinning these observations remain to be elucidated, these reports suggest that an extensive investigation into the functional significance of plectin and CSP to immunogenicity is warranted.

The dynamic interactions between cancer cells and the tumor microenvironment play a critical role in cancer development and progression [91]. For example, in acute myeloid leukemia (AML), the interaction between leukemic cells and the bone marrow (BM) microenvironment has been suggested to drive disease progression and chemoresistance. Interestingly, in a genome-wide analysis of bone marrow mesenchymal stromal cells (BM-MSCs), a major component of the BM niche, derived from AML patients, found only a non-synonymous alteration in the plectin gene (R1801Q) to be mutated at diagnosis, complete remission, and relapse of the same patient [93]. Moreover, plectin was revealed to be significantly overexpressed in AML BM-MSCs compared to healthy donor BM-MSC, suggesting a role for plectin in the AML microenvironment [93]. Future studies in larger cohorts are required to unravel if and how plectin influences the stroma–leukemia interaction. An analysis of the effect of 54 independent ascites samples from patients from a chemonaïve ovarian cancer cell line, OV-90, demonstrated that ascites fluid could have varying effects, from strongly stimulating to inhibiting, on invasive capacity [94]. Interestingly, plectin mRNA expression was significantly increased in OV-90 cells exposed to the acellular fraction of ascites fluid, with inhibitory effects on invasion [94]. However, with a Pearson correlation of 0.322 (*p* value = 0.19) between plectin genes expression and the impact of ascites fluid on invasion, more extensive validation is warranted to understand the complex role of plectin in OC.

### 3.4. CSP Regulates Malignant Hallmarks

CSP has emerged as a cancer stem cell (CSC) marker and mitigator of malignant phenotypes (Figure 3AIV). CSCs are a subpopulation of cells within a tumor, characterized by their capability of self-renewal, differentiation, and tumorigenicity. They have been implicated in playing a critical role in cancer relapse and metastasis [95]. Previously, plectin was associated with cancer stem cell marker OCT4A in ovarian cancer [96,97]. Recently, Raymond et al. applied an unbiased peptoid combinatorial cell screen to identify CSP as a selective CSC cell-surface marker [16]. Strikingly, isolated CSP-positive NSCLC cells showed elevated expression of plectin and key stem cell markers (ALDH1A3, SOX2, and CD44) and increased clonogenicity compared to CSP-negative cells from the same cell line [16]. In PDAC cells, incubating plectin-rich exosomes with CSP-negative cells conferred CSP expression and increased cancer cell migration and invasion capacity [12]. Similarly, the overexpression of plectin-1a and -1f isoforms was also revealed to induce CSP expression on cancer cells previously devoid of CSP [12]. Moreover, a recent study by Perez et al. implemented a pharmacologic approach to investigating CSP’s function in ovarian cancer [98]. A first-in-class CSP-targeting monoclonal antibody, 1H11, was shown to internalize without interacting with intracellular plectin, thus inducing CSP-specific targeted consequences. Inhibiting CSP caused G0/G1 arrest, decreased cell migration, downregulated the JAK2-STAT3 pathway, and suppressed tumor growth in vivo [1]. These reports emphasize the importance of further elucidating CSP’s mechanism of action and evaluating its therapeutic potential across different cancers.

## 4. Clinical Utility of Plectin

### 4.1. Plectin as a Risk Factor

Testicular germ cell tumor (TGCT), the most frequent testicular cancer, has an estimated heritability of 37% to 49%, the third-highest among all cancers [99]. Thus, Paumard-Hernández et al. aimed to identify novel susceptibility genes involved in TGCT by performing whole-exome sequencing on 19 Spanish familial TGCT cases and evaluating variants against data from TCGA and the Exome Aggregation Consortium (ExAC) databases [100]. A plectin variant (p.Arg433Gln) was revealed to be significantly associated with a higher risk of TGCT [100]. Moreover, an analysis of copy number variation in ESCC patients and cancer-free individuals from Southwest China revealed a copy number gain of PLEC as conferring increased ESCC risk [101]. Future studies with larger sample sizes would further our understanding of plectin’s clinical significance for risk stratification.

### 4.2. Plectin as a Prognostic Indicator

Plectin IHC analyses demonstrated that high plectin expression is a significant indicator of worse overall survival in HNSCC and non-metastatic OSCC patients [10,37]. In lung cancer, high plectin mRNA expression is significantly associated with worse overall patient survival (adenocarcinoma and squamous cell carcinoma; hazard ratio (HR) = 1.46, *p* = 7.9 × 10^−7^) [16]. Further analysis revealed that this held true regardless of gender and smoking status [16]. Interestingly, plectin expression showed higher prognostic significance in non-smokers (HR = 0.75, *p* = 2.8 × 10^−5^) than smokers (HR = 1.9, *p* = 4.8 × 10^−4^) [16]. Stratifying patients by histology revealed high plectin mRNA expression correlated with worse overall survival in adenocarcinoma patients (HR = 2.2, *p* = 5.5 × 10^−11^) but with better overall survival in squamous carcinoma patients (HR = 0.75, *p* = 0.044) [16]. In contrast, patient survival analysis in OSCC did not show a significant association between high plectin IHC staining and overall survival or progression-free survival [37].

We performed an expanded study to evaluate the prognostic utility of plectin mRNA expression on overall patient survival across different cancers using the KM plotter database (Figure 4) [102]. Strikingly, high plectin mRNA expression was significantly associated with worse overall survival in multiple cancers, including PDAC, lung adenocarcinoma, and HNSCC, for which plectin has been previously revealed as a protumorigenic regulator [10,12,16]. In contrast, low plectin expression was an indicator of worse overall survival in sarcoma, thymoma, pheochromocytoma, and paraganglioma, suggesting that plectin’s role in cancer could be tissue- or context-dependent. Moreover, the revelation of plectin’s prognostic significance in several CSP-positive cancers, including PDAC, ovarian, and lung carcinoma, gives rise to an intriguing hypothesis that plectin’s localization may drive its prognostic importance [15,16]. These results highlight the need to delineate how intracellular plectin and CSP independently drive oncogenesis to inform future diagnostic and treatment strategies. Additional investigations could also evaluate how plectin and CSP function may differ according to cancer subtypes and commonly harbored mutations. Overall, these insights strengthen the notion that plectin plays a critical role in cancer tumorigenesis.

### 4.3. Plectin as a Diagnostic Biomarker

More than 80% of pancreatic cancer patients are initially diagnosed with advanced-stage disease, for which the 5-year survival rate is < 5%, underscoring the need for improving diagnostic modalities such as endoscopic ultrasound-guided fine-needle aspiration (EUS-FNA) [103]. As previously described, plectin is an established and widely validated biomarker for PDAC [14]. Thus, a multicenter study evaluated plectin IHC as a biomarker of malignant risk for precursor legions to PDAC and IPMNs in surgical and EUS-FNA samples [104]. In surgical specimens, plectin expression was heterogeneous in IPMNs; however, percent staining increased from low-grade dysplasia to high-grade dysplasia [104]. Moreover, a significant increase in plectin staining between in situ IPMN and patient-matched invasive PDAC was reported [104]. Plectin IHC differentiated IPMNs without invasive PDAC from those with an invasive PDAC component with a specificity of 85% and sensitivity of 75% [104]. Overall, plectin IHC was determined to have insufficient accuracy in differentiating low-grade dysplasia IPMNs (LGD IPMNs) and high-grade dysplasia IPMNs (HGD IPMNs); however, it did demonstrate a negative predictive value of 72% [104]. Similarly, plectin IHC on cytological samples could not significantly differentiate between LGD and HGD IPMNs [104]. This report by Moris et al. is in contrast to an earlier study by Bausch et al. that reported plectin IHC distinguished malignant IPMNs (HGD IPMNs and invasive PDAC) from benign IPMNs (low and moderate dysplasia) with 83% specificity and 84% sensitivity [14]. This discrepancy could be due to differences in how specimens were segregated, patient sample size (Moris et al.: 39 benign samples; Bausch et al.: 6 benign samples), IHC scoring system (Moris et al.: semiquantitative; Bausch et al.: dichotomized), and technical variations in specimen handling and observer scoring [14,104]. However, both studies did coincide in reporting high uniform plectin expression in invasive PDAC samples and HGD with an invasive PDAC component [14,104].

Overall, conventional EUS-FNA cytology is hindered by relatively low diagnostic accuracy (59%) due to low sensitivity, gastrointestinal contamination, and interobserver variability [105]. Cyst fluid analysis for biomarkers of malignancy could help improve the overall accuracy of diagnosis of IPMNs. Stinkingly, Bausch et al. reported that plectin could be immunoprecipitated from cyst fluid samples of malignant but not benign IPMNs [14]. These results suggest that plectin could serve as a diagnostic marker in pancreatic cyst fluid samples. Moreover, one of the hallmarks of PDAC found in over 90% of tumors is the point mutation of KRAS, and thus, KRAS mutation analysis has been shown to increase the diagnostic yield of EUS-FNA for PDAC tissues [106]. Therefore, Park et al. investigated the diagnostic utility of incorporating plectin staining of EUS-FNA tissues in addition to KRAS mutation analysis [107]. Consistent with previous reports, Park et al. reported that normal pancreas tissues showed none-to-minimal plectin expression while 100% of the PDAC tissues analyzed did [107]. Furthermore, the combination of plectin staining with standard cytology and KRAS mutation analysis increased the sensitivity (96%), specificity (93%), and accuracy (95%) of pathologic diagnosis of PDAC in EUS-FNA samples compared to conventional cytology (81%, 80%, and 79%, respectively) [107]. Further studies into the diagnostic role of plectin staining in differentiating early-stage PDAC lesions from benign tissue with EUS-FNA specimens could further inform early diagnostic strategies.

In lieu of traditional tumor tissue biopsy, which is invasive and only provides a static representation of the tumor, liquid biopsies are non-invasive and can present real-time insights throughout the course of disease [108]. Hence, Song et al. evaluated plectin as a marker of circulating tumor cells (CTCs) from portal vein and peripheral blood samples of early-stage PDAC patients [109]. They were able to identify plectin-positive CTCs in 43.8% and 50% of portal blood and peripheral blood samples, respectively. Moreover, no plectin-positive CTCs were detected in samples from healthy volunteers [109]. However, the number of plectin-positive CTCs did not relate to disease staging, nor did it show a significant association with overall survival [109]. Thus, expanded analysis of patients at varying PDAC stages could elucidate if and how plectin-positive CTCs change during PDAC carcinogenesis. Alternatively, plectin-rich exosomes are detectable from the serum of PDAC tumor-bearing mice, opening the possibility of plectin as a serum marker [12]. These observations underscore the need for a more detailed evaluation of plectin as a powerful indicator for early screening of PDAC.

## 5. Plectin-Targeting Imaging Agents and Therapeutics

Precision medicine holds the promise of improving patient care by administering therapies tailored to a patient’s unique genetic and proteomic profile. To maximize precision medicine’s therapeutic impact, there is a need to identify unique molecular targets and biomarkers that can inform patient stratification and predict efficacy. Plectin’s protumorigenic role and unique cell surface mislocalization in several cancers make it an ideal candidate for targeted imaging and therapeutic strategies (Figure 5).

### 5.1. Imaging Agents

Targeted molecular imaging holds great potential to enable earlier diagnosis and detection of cancer. CSP’s bioavailable, abundant, and cancer-specific expression makes it an ideal cell-surface antigen for targeted strategies in cancer [13,32]. In Kelly et al.’s seminal study, they identified KTLLPTP as a novel plectin-targeting peptide (PTP) and demonstrated that PTP magnetofluorescent nanoparticles (NPs) accumulated in PDAC tumors of a genetically engineered mouse model but not to normal pancreatic tissue [13]. In a follow-up study, the Kelly group generated a CSP-targeting imaging agent for single-photon emission/CT (SPECT) by conjugating PTPs (KTLLPTP) to ^111^In (^111^In-PTP) [32]. Administration of ^111^In-PTP to three different orthotopic xenograft mouse models of PDAC resulted in significant accumulation at tumors and peritoneal metastases, visualized by SPECT/CT imaging [32]. Biodistribution studies of harvested organs and tumors revealed 1.9- to 2.9-fold and 1.7-fold increases in ^111^In-PTP uptake in tumors and liver metastases compared to normal pancreas and liver, respectively [32]. Since then, PTP has been leveraged in a multitude of other studies to guide imaging agents and drug delivery systems to CSP-positive tumor tissue [18,19,20,21,22,23,24]. For example, using multiphoton microscopy, PTP-lipid microbubbles were shown to bind selectively to PDAC cells (PANC-1 and MIAPaCa-2) but not to healthy cells (hTERT-HPNE) [110,111]. In addition to CSP’s abundance in PDAC, integrins have also been reported as biomarkers of pancreatic cancer and tumor vasculature [13,112,113]. Consequently, a bispecific molecular probe targeting both CSP (via PTP) and integrins for MRI/near-infrared imaging (MRI/NIRF) was shown to bind specifically and with higher efficiency to PDAC cells in vitro and in vivo [114]. Moreover, the bispecific molecular probe successfully aided the guidance of NIRF surgery in an orthotopic PDAC xenograft model [114]. CSP could, therefore, serve as a valuable target to amplify current imaging strategies.

Iron oxide nanoparticles are highly attractive imaging agents due to their adaptability for use in MRI, magnetic particle imaging (MPI), and magnetic hyperthermia [115]. Wang et al. generated bovine serum albumin superparamagnetic iron oxide nanoparticles (BSA·SPIONs), which they conjugated to a near infra-red fluorescent dye (Cy5) and anti-plectin monoclonal antibodies [22]. They demonstrated that CSP-targeting BSA·SPIONs selectively bound to malignant Panc-1 cells in vitro [22]. Since then, Chen et al. also developed a CSP-targeting iron oxide nanosystem with a near-infrared fluorescent dye (Cy7), which they detected using MRI and optical imaging [23]. The CSP-targeting nanoparticles showed high accumulation in PDAC cell lines (MIAPaCa-2 and XPA-1) but not in non-malignant MIN6 cells [23]. In vivo, using a PDAC orthotopic xenograft model, their dual-functional CSP-targeting probes showed greater uptake to PDAC tumors but not in normal pancreatic, liver, or kidney tissues [23]. These results further demonstrate the potential value of CSP-targeting nanoparticles for cancer detection.

### 5.2. Plectin-Guided Drug Delivery

#### 5.2.1. Polymeric Nanoparticles

Cancer cells have been characterized to have elevated levels of endogenous reactive oxygen species (ROS); however, excessive ROS accumulation can result in cell senescence and death [116]. Consequently, cancer cells are more sensitive to drugs that inflict ROS-mediated anti-cancer effects [116]. Quinazolinedione-based compounds (QDs), such as QD242, have been shown to induce cytotoxicity in cancer cells by inducing ROS generation [117]. To this end, PTPs (KTLLPTPC) have been successfully implemented to guide the delivery of QD242-encapsulated polymeric nanoparticles to CSP-positive PDAC cells, MIAPaCa-2, and trigger internalization [19]. Targeted QD242-NPs demonstrated a significantly higher cytotoxic effect than non-targeted NPs in vitro; however, further studies are required to evaluate the specificity and efficacy of CSP-targeting QD-NPs in vivo [19].

The tumor stroma plays a critical role in tumorigenesis, metastasis, and therapeutic response. The failure of clinical trials focused on stromal depletion has underscored the stroma’s complex role as both an obstacle for therapy and a safeguard against metastasis [118]. Thus, recent therapeutic approaches have focused on normalizing the stroma to increase drug delivery and disrupt the cross-talk between cancer-associated fibroblasts and tumor cells. To this end, PTPs (KTLLPTPC) have been used to improve the accumulation of an MMP-2-responsive nanopolyplex in PDAC models [21]. Li et al. have shown that the delivery and efficiency of a newly created copolymer designed to co-deliver a TGF-β inhibitor (LY2109761) and a chemotherapy agent (CPI-613) was greatly enhanced by the incorporation of PTP [21]. The CSP-targeted polyplex shown greater accumulation in tumors of both subcutaneous and orthotopic PDAC mouse models. As a result, treatment with the CSP-targeted dual-drug-loaded polyplex significantly inhibited tumor growth, decreased collagen and α-SMA expression, and increased cancer cell death above that seen from treatment with the free drug alone and untargeted polyplex [21]. These results demonstrate CSP’s utility in guiding therapeutic strategies that dual-target cancer cells and the stroma.

miRs have been revealed to play multifaceted roles in cancer as tumor suppressors or oncogenes, prompting increased interest in utilizing miRs as therapeutic tools and targets [20,119]. However, miR-based therapies are hindered by low membrane penetrability, poor biological compatibility, and off-target effects [120]. To address these limitations, a miR delivery system that utilizes PTPs (KTLLPTP) to specifically and efficiently guide treatment to cancer cells was developed. Chen et al. describe the generation of a chimeric peptide consisting of PTP that can capture miR through electrostatic interactions and self-assemble into nanoparticles [121]. This system has been used in PDAC models to deliver miR-9 and miR-22, both regulators of autophagy and sensitivity to doxorubicin, resulting in enhanced delivery and penetrability in vitro and in vivo [20]. In both instances, dual treatment of PTP-miR nanoparticles and doxorubicin resulted in greater tumor volume reduction than doxorubicin alone [20]. These studies demonstrate the therapeutic utility of CSP-targeting nanoparticles in increasing the efficacy of miR-based therapy in cancer.

#### 5.2.2. Gold Nanoparticles

Out of the array of nanoparticles available for drug delivery, gold nanoparticles (GNPs) stand out for their chemical stability, biocompatibility, and optical properties. Moreover, they can be easily synthesized and conjugated to various molecules [122]. Pal et al. reported on the generation of GNPs whose surface is modified with PTPs (KTLLPTPYC) and conjugated to gemcitabine (GNP-Gem) [24]. Evaluation of in vitro efficacy found that GNP-Gem induced higher cytotoxicity in two PDAC cell lines, AsPC-1 and PANC-1. Interestingly, even without conjugation with gemcitabine, treatment with plectin-targeting GNPs also decreased cell viability, suggesting that targeting CSP could hold therapeutic potential. In vivo, using a PDAC orthotopic xenograft model, treatment with GNP-Gem resulted in an enhanced reduction in tumor volume and Ki67+ proliferating cells compared to treatment with gemcitabine alone. Moreover, surface modification with PTPs resulted in GNPs accumulating in tumor tissues and not adjacent normal tissues [24]. These results highlight how targeting CSP can efficiently steer drug delivery systems to PDAC tissues while evading normal tissues.

#### 5.2.3. Targeted Adeno-Associated Virus (AAV) Particles

Engineered AAV vectors are a promising gene delivery system due to their stability, strong safety profile, production scalability, and vector adaptability [123]. Moreover, the integration of a cancer-specific ligand on the surface of the viral capsid holds promise for the development of novel diagnostic and treatment modalities. As a proof of principle, Konkalmatt et al. developed a CSP-targeting AAV2 vector that is selectively bound to CSP, resulting in the preferential transduction of PDAC cells over non-neoplastic cells in vitro and in vivo (Figure 1) [18]. The loop IV region of the AAV2 capsid was replaced with a PTP (KTLLPTP) without sacrificing viral titer. In vitro, the CSP-targeting AAV2 vectors preferentially transduced multiple CSP-positive PDAC cell lines (PANC-1, MIAPaCa-2, HPAC, MPanc-96, and BxPc3) up to 30-fold higher than control capsid. Using a xenograft subcutaneous model, CSP-targeting AAV2 showed selective accumulation at tumors while control capsids preferentially transduced hepatocytes, its natural tropism (Figure 1) [18]. This AAV delivery system demonstrates the potential of CSP-targeting for the use of gene therapy in CSP-positive tumors.

#### 5.2.4. Targeted Liposomes

Liposome-based drug delivery has been widely exploited due to its low immunogenicity, favorable pharmacokinetic and pharmacodynamic profiles, and enhanced efficacy and safety [124]. Leveraging that ovarian cancer cells are CSP-positive, CSP-targeted liposomes have been successfully implemented to increase the drug payload of PARP inhibitor AZ7379 to ovarian cancer cells (OVCAR8 and SKOV3) [15]. In vivo, using both subcutaneous and intraperitoneal mouse models of ovarian cancer, treatment with drug-loaded CSP-targeting liposomes resulted in a significant increase in the inhibition of PAR activity and tumor growth compared to systemic and untargeted liposomal treatment [15]. Dasa et al. demonstrate that CSP-targeting therapies help to increase efficacy in other CSP-positive tumors beyond PDAC.

#### 5.2.5. Natural Protein Drug Delivery Systems

Although synthetic drug delivery systems such as liposomes are easily adaptable and readily able to carry drugs using their hydrophobic core, they are limited by clearance through the reticuloendothelial system in the liver and spleen [122]. Moreover, challenges with controlled drug release raise questions of efficacy and safety [122]. An alternative approach is to utilize natural biological compartments as drug delivery vehicles. Yuan et al. have described the use of GroEL, a chaperone with two hydrophobic cavities, for delivery of doxorubicin to CSP-positive pancreatic and breast cancer tumors [28]. GroEL was characterized to bind specifically to CSP on pancreatic cancer cells (Panc-1) and breast cancer cells (MDA-MD-231) but showed no binding to normal epithelial cells (HPNEs) and normal keratinocytes (HaCaTs) [28]. In addition to immunoprecipitation experiments, Yuan et al. demonstrated that GroEL’s binding to CSP was markedly reduced by competitive inhibition with anti-plectin antibodies [28]. GroEL is an ATPase; thus, upon excitation, it undergoes a conformational change to release its substrate. Leveraging that the tumor microenvironment has elevated levels of ATP, Yuan et al. loaded GroEL with doxorubicin (GroEL-Dox) and evaluated its ability to effectively and selectively deliver doxorubicin to tumors. In both pancreatic and breast cancer xenograft models, GroEL-Dox significantly inhibited tumor growth without major toxic effects compared to doxorubicin alone [28]. Given CSP’s ability to effectively steer targeted therapies, identifying other cage-like interacting partners could prove beneficial.

### 5.3. Plectin and CSP-Targeted Therapeutics

Plectin is a potent mitigator of multiple tumor activities, including cell survival, proliferation, migration, and invasion. Knockout experiments have emphasized the profound anti-tumor effects of disrupting plectin function [10,11,12]. Thus, exciting reports have emerged that expand and mobilize our understanding of plectin by investigating the therapeutic potential of its direct targeting. This new avenue of research is widening our anti-cancer repertoire and utilizing a pharmacological approach to interrogate plectin’s and CSP’s function in cancer.

#### 5.3.1. Metallodrugs

The application of metallodrugs for cancer treatment is of great interest due to their ease of chemical modification and wide-spectrum mechanisms of action. However, the adverse side effects and the emergence of drug resistance limit the use of platinum first-line cancer drugs such as cisplatin, carboplatin, and oxaliplatin [125]. Alternatively, plecstatin-1, an organoruthenium drug candidate that selectively targets plectin and CSP, has been identified as a promising anti-cancer strategy [25,26,27]. In colon cancer cells (HCT116), administration of plecstatin-1 induced G0/G1 arrest [26]. Furthermore, in both HCT116 and breast cancer cell line MDA-MB-231, plecstatin-1 reduced mitochondrial membrane potential and increased levels of ROS [25,26,27]. In a colorectal tumor spheroid model, treatment with plecstatin-1 resulted in reduced spheroid growth, disruption of F-actin and α-tubulin networks, and increased phosphorylation of stress protein eIF2α [25]. Consistent with these observations, a Gene Ontology (GO) term analysis of plectin-targeted cells revealed the downregulation of mitochondrial proteins and the upregulation of proteins relating to cytoskeletal organization [26,27]. Moreover, plectin targeting with plecstatin-1 reduced tumor growth in B16 melanoma and CT-26 colon tumor mouse models [26,126]. Interestingly, an NCI-60 screen revealed plecstatin-1 as having higher potency than cisplatin alone [26]. While pharmacokinetic and pharmacodynamic studies are required to evaluate the clinical translatability of plecstatin-1, its potent anti-cancer effects call attention to the selective targeting of plectin as a robust anti-cancer strategy.

#### 5.3.2. Monoclonal Antibody

Monoclonal antibodies are ideal for increasing treatment tolerability and outcomes due to their high affinity, specificity, and favorable pharmacokinetics. Currently, there are only about two dozen cell-surface proteins targeted by approved therapeutic antibodies, underscoring the need to widen our cell-surface antigen repertoire [127]. In a recent study featured in this *Cells* Special Issue, Perez et al. developed a first-in-class anti-CSP monoclonal antibody, 1H11, that binds to CSP-positive cells with high affinity and specificity, after which it triggers rapid internalization and sequestering of 1H11 to vesicles [98]. In vitro, 1H11 selectively induced cytotoxicity, G0/G1 arrest, and decreased cell migration in CSP-positive ovarian cancer cells, OVCAR8 and SKOV3, but not healthy fallopian tube cells (FT132) [98]. These observations were associated with inhibition of the JAK2-STAT3 pathway, upregulation of cyclin-dependent kinase inhibitors, p21 and p27, and differential expression of EMT markers E-cadherin and vimentin. Consistently, in vivo treatment with 1H11 induced sustained tumor growth inhibition and resulted in 30% tumor necrosis compared to the IgG control group. IHC analysis of tumor tissue revealed 1H11 inhibited cell proliferation, resulting in decreased Ki67 and increased p21 staining [98]. CSP inhibition recapitulated the anti-cancer effects seen from plectin knockout studies, seemingly implicating CSP as a potent mediator of plectin’s protumorigenic function. Importantly, 1H11 did not induce major toxic effects. Strikingly, 1H11 was shown to synergize with current mainstay therapies, including cisplatin, olaparib, and doxorubicin. In vivo, dual treatment of 1H11 and cisplatin resulted in 60% greater tumor growth inhibition than cisplatin alone [98]. Taken together, Perez et al. are the first to expand on and marshal our understanding of CSP as a valuable therapeutic target. Their work opens the door for future avenues of investigation into CSP and demonstrates the utility of noncanonical cell surface proteins to be leveraged for anti-cancer interventions.

## 6. Role of Plectin and CSP in Drug Sensitivity and Resistance

CSP has recently been revealed as a mitigator of drug sensitivity. The pharmacologic inhibition of CSP with a monoclonal antibody has been shown to increase the potency of cisplatin, doxorubicin, and olaparib in ovarian cancer cells [98]. In vivo, combinational treatment with the anti-CSP antibody and cisplatin resulted in enhanced suppression of tumor growth by up to 60% more than cisplatin alone [98]. This observation emphasizes the potential of CSP inhibition for integration with current mainstay anti-cancer treatments.

Plectin has also been implicated in modulating the sensitivity of small molecule drugs. For example, in vitro, lower plectin expression correlated with higher drug sensitivity to sorafenib, a tyrosine kinase inhibitor shown to improve HCC patient survival [90]. Vitamin D has been demonstrated to elicit anti-cancer effects such as inhibiting cancer cell proliferation, reducing metastatic potential, and inducing cell death [128,129]. Using human colon cancer cell line SW480-ADH, treatment with a vitamin D metabolite, 1α,25(OH)_2_D_3_, increased plectin expression, inhibited cell proliferation, and induced epithelial differentiation [129]. While these observations have yet to be validated in vivo, they provide a basis for further evaluation of plectin as a marker for and a potential modulator of drug sensitivity. Interestingly in fibrosarcoma, proteomic analysis revealed that treatment with a vascular disrupting agent, combretastatin A4-phosphate, reduced plectin expression compared to untreated tumors [44]. IHC staining demonstrated that plectin only localized to viable tumor regions; thus, further study is warranted to explore if decreased plectin expression is due to increased tumor necrosis or if plectin expression plays a functional role in cell survival after treatment with a vascular disrupting agent [44]. Moreover, proteomic analysis with ovarian cancer SKOV3 cells revealed reduced plectin expression after treatment with immunotherapies, protein aggregate magnesium-ammonium phospholinoleate-palmitoleate anhydride (P-MAPA), and interleukin-12 as single agents or in combination [130]. The clinical utility of this association remains unknown.

Several studies have suggested a link between plectin and chemoresistance. Proteomic analysis of non-adherent cancer cells derived from ascites of advanced-stage ovarian cancer patients found that plectin expression was significantly higher in samples from chemonaïve patients at diagnosis compared to patients with recurrent disease after chemotherapy [97,131]. It is suggested that this differential expression pattern is due, in part, to recurrent disease samples having a free-floating phenotype with a concomitant decrease in cytoskeletal adhesion proteins and a change towards a mesenchymal program [97,131]. In vitro, treatment with paclitaxel or cisplatin was shown to enhance plectin expression across different ovarian cancer cell lines (e.g., HEY, SKOV3, and OVCAR5) [96]. In an intraperitoneal murine model, mice with recurrent disease after initial paclitaxel treatment showed significantly higher plectin expression than mice with non-recurrent paclitaxel treatment and untreated controls [97]. Further evaluation of the relationship between plectin expression and disease recurrence after chemotherapy could inform the integration of plectin-targeting therapeutics in mainstay treatment regimes.

## 7. Concluding Remarks

The wave of new targeted therapeutics reflects the clinical need for more effective, less systemically toxic treatment options to treat cancer, which remains a leading global cause of death. However, targeted therapies are plagued by the target’s non-selective nature, leading to toxicity, variable response rates, and the development of resistance. As a result, there is an urgent demand to widen our portfolio of bioavailable therapeutic targets and expand our understanding of optimal combinatorial strategies. Contrary to the influx of targets identified by genetic and sequencing strategies, plectin is a notable target identified by an alternative drug discovery approach that emphasizes cell-surface localization. Over the past decade, since first being revealed as a pancreatic cancer biomarker, our understanding of plectin has dramatically evolved, expanding its application as a potential biomarker to multiple cancers and leading to the development of plectin and CSP-targeted imaging and drug delivery modalities, elucidation of its various roles in cancer progression, and the design of anti-plectin therapies with potent anti-cancer effects.

Plectin demonstrates increased expression in tumor tissue compared to normal tissue across multiple cancers, including pancreatic, ovarian, prostate, lung, and head and neck carcinoma, among others [10,11,15,17,32]. Further characterization, particularly among cancers with a high differential expression that lack diagnostic biomarkers, could prove to markedly impact early detection rates, leading to a more favorable prognosis. Interestingly, it is not only plectin’s expression or mutation status but also localization that is altered in cancer. Normally cytosolic, plectin is detected on the outside cell surface of tumor cells. The mechanism inducing its mislocalization, CSP’s cell surface binding partners, and a structural comparison of CSP and intracellular plectin remain to be investigated and could be the focus of future studies. Excitingly, CSP’s cell surface localization has been successfully exploited to reach tumors that respond well to highly toxic, broad-spectrum drugs that are poorly tolerated systemically. Combining a non-specific cytotoxic drug with a CSP-targeted moiety dramatically increases the specificity for tumor tissue over normal tissue (for example, Figure 1). Conjugating CSP-targeting peptides to the outside of drug-carrying nanoparticles of various kinds has been shown to increase the payload delivery to tumor tissue [19,20,21,24]. Work by Perez et al. with a novel CSP-targeting monoclonal antibody that is rapidly internalized upon binding also presents an enticing potential for an antibody–drug conjugate system, whereby similar anti-tumor effects could be produced with a cytotoxic drug without encapsulation [98]. Moreover, plectin expression levels not only serve as a biomarker and tumor target for drug systems but correlate with a worse prognosis, suggesting a role in tumorigenesis.

Indeed, an exploration into the function of the plectin mislocalized to the cell surface has demonstrated a role in many tumor processes, including cell cycle, migration, and immune escape, pointing to the potential of anti-CSP antibody therapy [16,25,26,98]. As presented in a companion research article in this *Cells* Special Issue, the investigation into the effects of a novel anti-CSP antibody in ovarian cancer revealed a dramatic decrease in tumor growth in ovarian cancer murine models and unraveled a wealth of anti-tumor mechanisms associated with anti-CSP treatment [98]. This work highlights the likely gain of function roles for cell surface plectin beyond its intercellular counterpart. It underscores the potential for new insights into carcinogenesis via the exploration of the contributions of intracellular plectin, cell surface plectin, or both in driving malignancy. As further strides in understanding the mechanisms involved in anti-CSP therapy are made, combination strategies can be designed for optimal disease control. Given that anti-plectin drugs have been shown to stimulate the immune system against cancer cells, such combination strategies may be able to induce more lasting remissions and have a more substantial impact on metastases than current treatment regimens [26,27,98]. Furthermore, due to plectin’s role in chemoresistance, combination strategies have a high potential for synergism, increasing the efficacy beyond a mere additive effect.

The numerous cancer hallmarks impacted by plectin and CSP highlight how the inhibition of one protein can have multifaceted, far-reaching effects. Overall, recent strides in understanding plectin’s functional roles in cancer, characterizing its diagnostic and prognostic implications, and realizing its therapeutic potential have generated an exciting momentum towards improving overall survival for many difficult-to-treat cancers.

## Figures and Tables

**Figure 1 cells-10-02246-f001:**
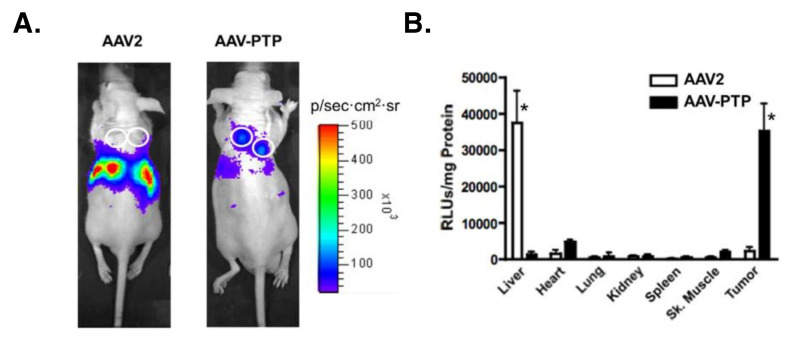
Adapted from Konkalmatt et al.’s study utilizing a CSP-targeting AAV2 vector (AAV-PTP) to selectively target gene delivery to PDAC tumors in vivo [18]. (**A**) Bioluminescence imaging of mice with subcutaneous PDAC tumors on day 14 after administration with AAV-PTP or wild-type AAV2 capsids harboring a luciferase reporter genome. AAV-PTP was predominantly localized to tumors with minimal activity in other regions. The white circles denote the site of tumors. (**B**) AAV-PTP’s selectivity for tumors was confirmed by measuring luciferase activity in major organs and tumors. * *p* < 0.05 comparing AAV2 and AAV-PTP in specified tissue.

**Figure 2 cells-10-02246-f002:**
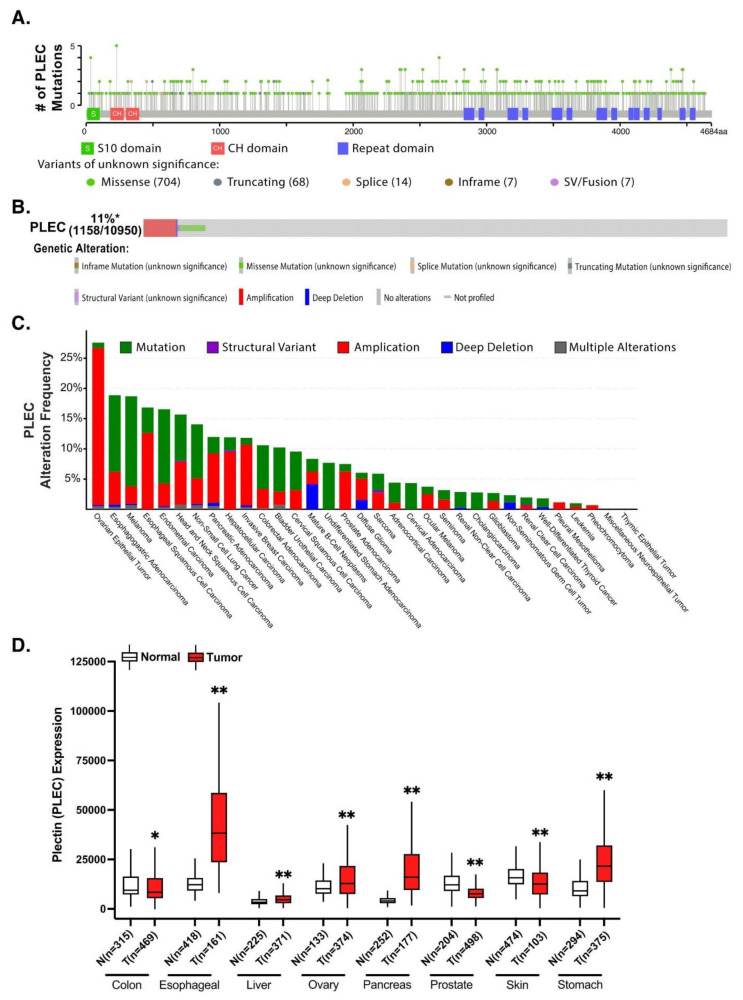
Plectin mutation analysis and expression across several cancers. (**A**) Plectin mutations from cBioPortal based on TCGA pan-cancer datasets (accessed 17 July 2021) [56,57]. (**B**) Oncoprint plot from cBioPortal demonstrating plectin is altered in 11% (1158/10,950) of all TCGA pan-cancer atlas studies. (**C**) The alteration frequency of plectin across different cancers from TCGA pan-cancer datasets using cBioPortal. (**D**) Plectin’s gene expression in tumor and normal tissue using the TNMPlot portal (accessed on 22 July 2021), which mines data from GEO, GTEx, TCGA, and TARGET databases [29]. N = normal tissue, T = tumor tissue, sample size of the cohort is specified below, * *p*-value < 0.01; ** *p*-value < 0.001; analyzed by Mann-Whitney U-test. Abbreviations, GEO: Gene Expression Omnibus, GTEx: Genotype-Tissue Expression, TARGET: Therapeutically Applicable Research to Generate Effective Treatments, TCGA: The Cancer Genome Atlas.

**Figure 3 cells-10-02246-f003:**
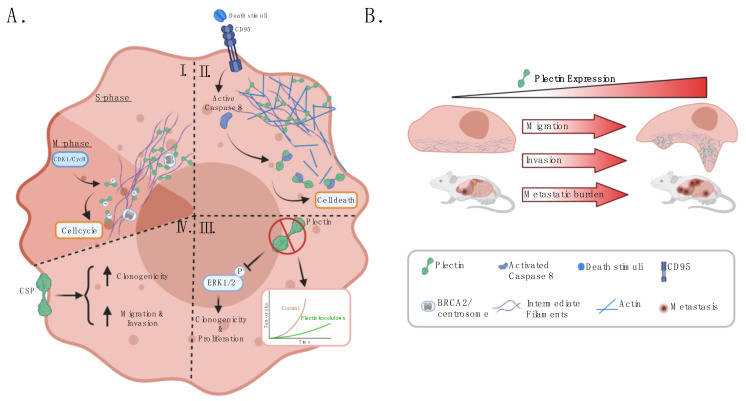
Plectin is a regulator of malignant hallmarks. (**AI**) During the S phase, plectin interacts with BRCA2 to localize the centrosome close to the nuclear membrane. During the early M phase, plectin is phosphorylated by CDK1/CycB, which prompts the disassembly of plectin’s crosslinking function with IFs and initiates centrosome movement, allowing nuclear duplication. (**AII**) Plectin binds and modulates the organization of actin. During CD95-mediated apoptosis, active caspase 8 cleaves plectin, triggering the disintegration of the stable actin cytoskeleton. (**AIII**) Plectin knockdown experiments have revealed that the ablation of plectin inhibits cell proliferation, clonogenicity, and tumor growth across different cancer models, including HNSCC, PDAC, NSCLC, and prostate cancer. In HNSCC, loss of plectin was shown to suppress the activation of ERK1/2, a critical proliferative regulator. (**AIV**) CSP-positive cells demonstrate increased cell clonogenicity, migration, and invasion compared to cells null for CSP expression. (**B**) Plectin is localized to podosomes and invadopodia, where it regulates and stabilizes the functional interaction between IFs and actin stress fibers. Loss of plectin disrupts the cytoskeleton, inhibits migration and invasion, and reduces metastatic burden in mouse models. Moreover, plectin is preferentially upregulated in aggressive cancer cells with high metastatic potential. Abbreviations, BRCA2: breast cancer susceptibility gene, CDK1/CycB: cyclin-dependent kinase 1/cyclin B, CSP: cancer-specific plectin, HNSCC: head and neck squamous cell carcinoma, PDAC: pancreatic ductal adenocarcinoma, NSCLC: non-small cell lung cancer.

**Figure 4 cells-10-02246-f004:**
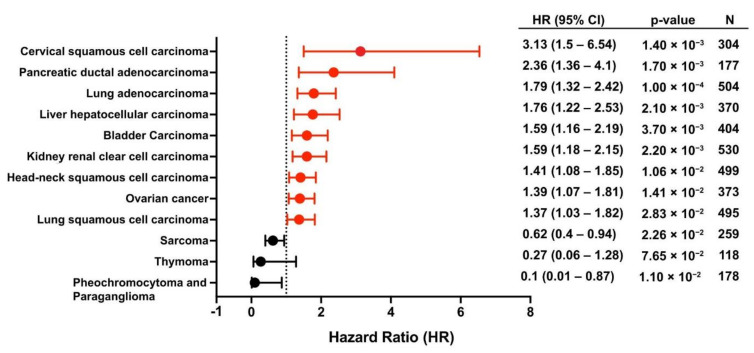
The prognostic value of plectin on overall patient survival across different cancer types was evaluated using the Kaplan–Meier plotter database’s pan-cancer option, consisting of RNA-seq data from TCGA repositories (accessed on 22 July 2021) [102]. The “auto-select” option was used to stratify patients into high- and low-expressing groups. The software calculated the HR, a measure of the relative risk of patients with high plectin mRNA expression compared to patients with low expression, and the 95% confidence intervals (CIs). The log-rank *p*-value and sample size (N) are specified. Increased risk: HR > 1 (red); reduced risk: HR < 1 (black); no difference = 1 (dashed line).

**Figure 5 cells-10-02246-f005:**
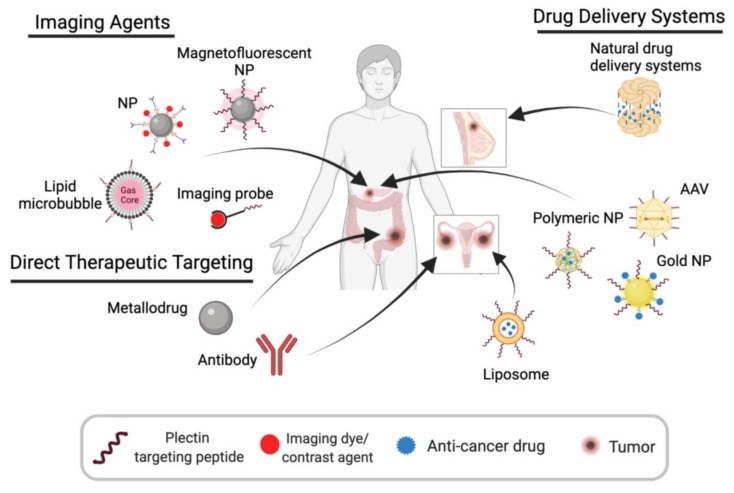
Plectin has been widely leveraged to enhance the specificity of imaging agents and the efficacy of therapeutic modalities. Plectin-targeting peptides (PTPs) or anti-plectin antibodies have been shown to increase the accumulation of nanoparticles (NPs), microbubbles, and imaging probes to PDAC cells and tumors. Similarly, PTPs have increased the drug payload of different drug delivery systems (e.g., liposomes, NPs, AAV) across various cancer models. Recent studies have revealed the anti-cancer effects of direct therapeutic targeting of plectin with metallodrugs in colon cancer and a monoclonal antibody in ovarian cancer, opening the door to a new class of plectin targeted therapy.

**Table 1 cells-10-02246-t001:** Plectin and its interacting partners regulate invasion and migration.

Protein Name	Cancer Type	Function of Protein	Effect of Interaction	Reference
Dlc1	Ovarian cancer	Tumor suppressor gene encoding a GTPase-activating protein	Plectin binds to transcriptional isoforms of Dlc1. Loss of Dlc1 increased focal adhesions and stress fiber formation.	[70]
KPNA2	Lung cancer	Imports proteins to the nucleus	An increase in complex formation is seen in lung adenocarcinoma cells with higher metastatic potential.	[86]
Periplakin	Breast cancer	Cytolinker plakin protein	Plectin isoforms plectin-1f and -1k interact with the N-terminus of periplakin at cell borders.	[69]
RON	Pancreatic cancer	Receptor tyrosine kinase	Upon binding its ligand, RON translocates to the cell surface and interacts with plectin at lamellipodia, disrupting the plectin–integrin β4 interaction via phosphorylation of PI3K, thus increasing cancer cell migration.	[68]
SNRPA1	Breast cancer	Mediator of alternative splicing	Exon 31 of plectin was identified as a SNRPA1 target, resulting in a rod domain-containing isoform that is upregulated in metastatic breast cancer tumors. Ablation of this plectin SNRPA1-mediated isoform resulted in reduced invasion and metastatic capacity.	[67]

Dlc1: deleted in liver cancer 1; KPNA2: karyopherin alpha; RON: recepteur d’origine nantais; SNRPA1: small nuclear ribonucleoprotein polypeptide.

## Data Availability

Not applicable.

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
