# Peer review of "Plectin in Cancer: From Biomarker to Therapeutic Target"

_cells, 2021, doi:10.3390/cells10092246_

Round 1

Reviewer 1 Report

In this Review, entitled “Plectin in Cancer: From Biomarker To Therapeutic Target”, Samantha M. Perez, Lindsey T. Brinton, and Kimberly A. Kelly provided a detailed discussion about the potential role of plectin as both biomarker and therapeutic target in multiple cancer types, which enable a more comprehensive understanding of their biological characteristics and targeting prospects. The strength of this review is that the summarization and analysis are presented in a clear and well-organized way, making the contents more informative and comparable. Moreover, the viewpoints expressed in each section include an outlook on future research rather than just a summary of the recent literature.

Although this Review is well written, several minor concerns should be addressed before accepting for publication in Cells.

  1. The cross-talks between plectin and immune system should be explicated more specifically.

  1. The foreseeable challenges to clinical implementation of the cancer-specific plectin (CSP)-targeted therapeutics remain unknown.

  1. In addition, this manuscript needs to be carefully proof-read.

Reviewer 2 Report

In the present manuscript, the authors review the role of plectin, especially cancer-specific plectin (CSP), in cancer. The paper is a detailed summary of the current knowledge of plectin’s function as an oncoprotein. One of the advantages of the paper is unquestionably the amount of the information; however, this also makes certain parts of the text relatively difficult to follow. Otherwise, the paper fits the scope of the Special Issue "Plectin in Health and Disease" nicely, and there is no doubt that it will be of interest to the readership of the journal. I would kindly ask the authors to address the following concerns:

Major comments:

  1. I appreciate the thorough work by the authors; however, it is imperative to improve the clarity of the review. I suggest that unnecessary details of the work that they are citing are excluded (e.g., numeri, p-values, Pearson correlations …). For example, the sentence (line 439-42): “Moreover, an analysis of copy number variation in 404 ESCC patients and 402 cancer-free individuals from Southwest China revealed a copy number gain of PLEC as conferring increased ESCC risk (Adjusted 441 OR = 3.725, 95% CI = 1.026-13.533, P = 0.046) [101].“ would be much easier to read, if the text highlighted in yellow would be omitted.
  2. One of the focal points of the review is the mislocalized cancer-specific plectin (CSP). Hence, it would be great if the authors would include a short paragraph where they would explain how (if) CSP differs from the “normal”, intracellular plectin, apart from the location? Is there a preferential isoform of plectin that comprises CSPs? What is the mechanism, by which plectin exits the cell? Is CSP bound to certain cytoskeletal elements at the outer leaflet of the plasmalemma (PM), or is maybe bound to the extracellular domains of known proteins in the PM? 

Minor corrections:

  1. Line 25: Plectin was originally isolated from rat glioma C6 cells, which was reported in the paper Pytela and Wiche (1980). Hence, it would be better, if the authors replace three decades with four decades.
  2. Lines 100-2: “Examination of plectin expression in PanINs showed that early lesions (PanIN I/II) had minimal expression (0 – 3.85%) while PanIN III lesions, considered carcinoma in situ, were 60% positive [32]“ Please explain, if the percentages denote overexpression of plectin or the percentage of plectin positive lesions. In the latter case, the authors should explain why normal expression of plectin, which is supposed to be expressed in virtually all cell types, is not detected/discerned (maybe, the authors meant differential expression?).
  3. Line 33: Here, intermediate filament acronym (IF) is used for the first time – it would be prudent to use the acronym throughout the text, instead of writing out the whole name and re-introducing it at line 303.
  4. Line 420: delete previous.
  5. Figure 4, please explain briefly what is hazard ratio.

Reviewer 3 Report

In their manuscript “Plectin in Cancer: From Biomarker To Therapeutic Target” Perez et al. reviewed plakin family member plectin as cancer marker and potential target for cancer treatment. The manuscript addresses a very interesting and topical field and given the amount of published works, similar review was surprisingly still missing. The manuscript is well structured and extremely well written, authors covered main papers reflecting the current state of plectin research providing thus an informative overview. In conclusion the review gives overall good impression, I have only few minor points concerning the current version, which may require the authors’ further consideration:

26-28: For clarity, please modify phrasing and join the text in italic into one sentence, e.g.: „It consists of (....) and a plakin repeat domain, giving a rise to at least 12 different isoforms by alternative splicing. As a result, plectin displays (...).

520: „lesions“ instead of „legions“

582: Missing „.“ at the end of the sentence.

Round 2

Reviewer 2 Report

In the revised manuscript, the authors have addressed my concerns and improved the manuscript. I have no other comments and suggest that the paper is accepted for publication